# Phase separation of a PKA type I regulatory subunit regulates β-cell function through cAMP compartmentalization

Ha Neul Lee[1], Julia C. Hardy[1,2], Emily H. Pool[1,3], Jin-Fan Zhang [1,2¤], Su Hyun Kim[1], William F. Buhl[1], Jessica G.H. Bruystens[1], Sohum Mehta[1], Susan S. Taylor[1,3], Jin Zhang[1,2,3,4]*

1 Department of Pharmacology, University of California, San Diego, La Jolla, California, United States of America, 2 Shu-Chien-Gene Lay Department of Bioengineering, University of California, San Diego, La Jolla, California, United States of America, 3 Department of Chemistry and Biochemistry, University of California, San Diego, La Jolla, California, United States of America, 4 Moores Cancer Center, University of California, San Diego, La Jolla, California, United States of America

◉ These authors contributed equally to this work.
¤ Current address: Department of Chemistry and Chemical Biology, Harvard University, Cambridge, Massachusetts, United States of America
* jzhang32@ucsd.edu

## Abstract

Cyclic adenosine monophosphate (cAMP), a ubiquitous second messenger, regulates a variety of cellular functions with high specificity. We previously showed that the type I regulatory subunit of cAMP-dependent protein kinase A (PKA), RIα, undergoes liquid–liquid phase separation (LLPS) to facilitate spatial compartmentalization of cAMP. However, how RIα LLPS regulates cellular function is largely unknown. Here, we identify the formation of RIα condensates in MIN6 β cells and reveal key roles for RIα LLPS in regulating β cell function. By combining CRISPR-based RIα knockout with an RIα mutant (Y122A) that exhibits defective cAMP-induced LLPS, we demonstrate that RIα LLPS drives cAMP compartmentalization to tune β cell $Ca^{2+}$ and cAMP oscillation frequency, control insulin secretion, regulate CREB-mediated gene expression and prevent uncontrolled proliferation. Our data establish the Y122A mutant as a selective molecular tool for studying RIα LLPS and expand our understanding of the functional impact of LLPS-driven protein assemblies.

## Introduction

3′,5′-cyclic adenosine monophosphate (cAMP) is a ubiquitous second messenger that precisely controls diverse cellular functions, including cell growth, proliferation, metabolism, survival, and mobility, by regulating key effector proteins such as protein kinase A (PKA) and exchange proteins activated by cAMP (Epac) [1,2]. In pancreatic β cells, cAMP is produced via classical GPCR signaling in response to hormones such as Glucagon-like peptide 1 (GLP-1) [3]. Glucose-induced oscillatory $Ca^{2+}$

**Data availability statement:** All relevant data are within the paper and its Supporting Information files.

**Funding:** This research was supported by National Institute of Diabetes and Digestive and Kidney (R01 DK073368 to J.Z.) and National Cancer Institute (NIH T32 CA009523 to H.N.L). The funders had no role in the study design, data collection and analysis, decision to publish, or preparation of the manuscript.

**Competing interests:** The authors have declared that no competing interests exist.

**Abbreviations:** ACs, adenylyl cyclases; AKAPs, A-kinase anchoring proteins; cAMP, cyclic adenosine monophosphate; CREB, cAMP-response element binding protein; C/Y, cyan-over-yellow; FRAP, fluorescence recovery after photobleaching; Fsk, Forskolin; GLP-1, glucagon-like peptide 1; HBSS, Hank's balanced salt solution; IBMX, 3-isobutyl-1-methylxanthine; LLPS, liquid–liquid phase separation; PBS, phosphate-buffered saline; PDEs, phosphodiesterases; PDL, poly-D-lysine; PKA, protein kinase A; PKA-C, PKA catalytic subunit; RIPA, radioimmunoprecipitation assay; RT-PCR, reverse transcription PCR; TBST, TBS-Tween20; TEA, tetraethylammonium chloride; WT, wildtype.

signals also drive cAMP dynamics through the actions of $Ca^{2+}$-dependent adenylyl cyclases (ACs) and phosphodiesterases (PDEs) [4]. cAMP signaling in turn modulates intracellular $Ca^{2+}$ levels, which regulate insulin secretion via $Ca^{2+}$-dependent exocytosis and control gene expression [4]. We previously demonstrated that cAMP, PKA, and $Ca^{2+}$ form a highly integrated, oscillatory signaling circuit in MIN6 β cells to regulate numerous cellular processes [5].

Like many other processes, cAMP-dependent regulation of specific β cell functions requires tight spatial control of cAMP accumulation. Both cAMP production by ACs and degradation by PDEs are important for shaping cAMP signaling compartments [6]. For example, proper induction of β cell $Ca^{2+}$ oscillations has been shown to depend on the highly compartmentalized interplay between $Ca^{2+}$ and cAMP oscillations mediated by ACs localized within signaling nanodomains [7]. Local degradation by PDEs has been proposed as especially critical to cAMP's exquisite specificity, yet recent studies have suggested that cells likely employ additional mechanisms to facilitate PDE-driven compartmentalization [6]. Recently, we discovered that biomolecular condensates formed through cAMP-dependent liquid–liquid phase separation (LLPS) of the PKA regulatory subunit RIα are critical for cAMP compartmentalization [8]. RIα condensates serve as a dynamic buffering system that allow PDEs to effectively degrade local cAMP and maintain cAMP microdomains. Notably, we observed the formation of RIα condensates in various cell types, such as cardiomyocytes and neurons, suggesting that RIα LLPS may be involved in regulating cellular function across many tissues [8]. However, it is still unclear how RIα biomolecular condensates impact cellular functions.

In this study, we show that RIα LLPS also occurs in MIN6 β cells. To investigate how RIα condensates regulate cellular function, we generated RIα-null MIN6 β cells and employed an RIα mutant [9,10] containing a single Y122A substitution, which exhibits defective cAMP-induced LLPS without other major signaling defects. Using these tools, we discovered that RIα LLPS plays a crucial role in $Ca^{2+}$ and cAMP oscillation, insulin secretion, gene transcription, and cell proliferation. Our work reveals cAMP compartmentalization through RIα LLPS is involved in essential cellular processes, highlighting the functional importance of biomolecular condensates.

## Results

### RIα undergoes LLPS under physiological conditions in MIN6 β cells

We employed MIN6 β cells as an experimental model, capitalizing on the demonstrated high spatial control of cAMP in these cells and their various cAMP-dependent cellular functions, including $Ca^{2+}$ oscillations [7]. To investigate whether RIα undergoes LLPS in MIN6 β cells at the endogenous level, we used CRISPR-based knock-in technology to tag endogenous RIα with GFP2 (S1A Fig) and performed fluorescence imaging. In the absence of any stimulation, RIα-GFP2 knockin MIN6 β cells exhibited clear fluorescent puncta with strikingly distinct morphologies: large, irregular puncta and small, circular puncta (Fig 1A). To examine whether these puncta exhibit any cAMP-dependent changes, we next stimulated RIα-GFP2 knockin MIN6 β cells with the AC activator Forskolin (Fsk) and the PDE inhibitor

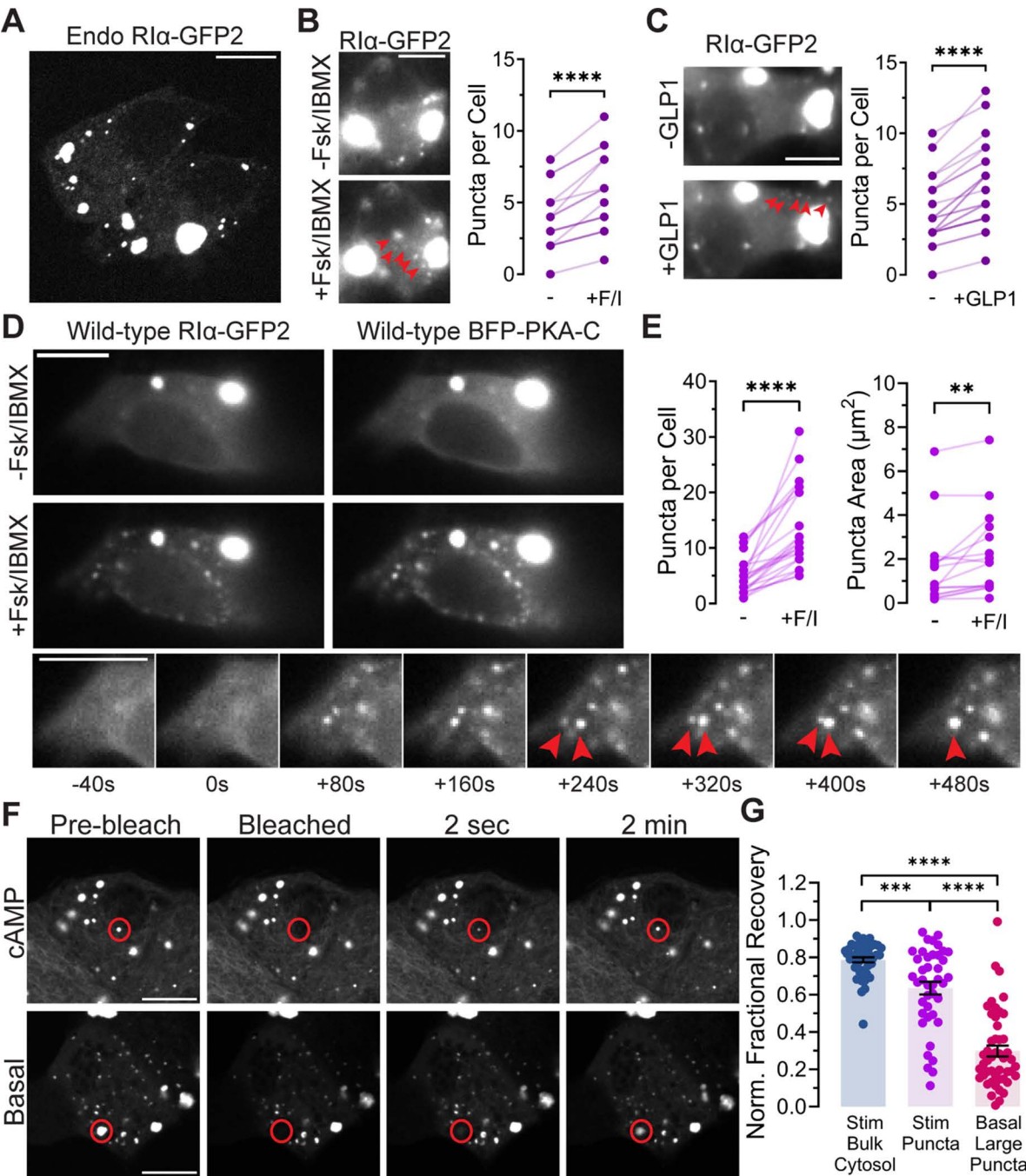

**Fig 1. RIα undergoes LLPS in MIN6 β cells.** **(A)** Confocal microscopy showing phase-separated bodies of endogenously GFP2-tagged RIα. **(B)** Representative fluorescence images **(left)** and quantification of endogenous RIα puncta per cell ($n = 16$) **(right)** in RIα-GFP2 knock-in MIN6 β cells before and after treatment with forskolin (Fsk, 50 μM) and IBMX (100 μM). ****$P < 0.0001$, paired, two-tailed Student t *test*. Arrowheads: newly formed puncta **(C)** Representative fluorescence images **(left)** and quantification of endogenous RIα puncta per cell ($n = 23$) **(right)** in RIα-GFP2 knock-in MIN6 β cells before and after treatment with GLP-1 (1 μM). ****$P < 0.0001$, paired, two-tailed Student t *test*. Arrowheads: newly formed puncta. **(D, E)** Representative fluorescence images and time-course images of RIα-GFP2 after Fsk/IBMX stimulation **(D)** and quantification of RIα puncta number per cell ($n = 22$ cells) **(E left)** and area (for pre-existing puncta) per cell ($n = 16$) **(E right)** in MIN6 β cells co-expressing RIα-GFP2 and TagBFP2-PKA-Cα before (−) and after (+) Fsk (50 μM) and IBMX (100 μM) stimulation. ****$P = 9.63 \times 10^{-7}$; **$P = 7.96 \times 10^{-3}$; paired, two-tailed Student $t$ test. Arrowheads indicate two puncta undergoing coalescence. **(F)** Representative fluorescence images of photobleaching and fluorescence recovery of basal and cAMP-stimulated RIα-GFP2 puncta in MIN6 β cells. **(G)** Quantification of the normalized recovery after photobleaching of stimulated bulk cytosol ($n = 44$ regions), stimulated

puncta ($n = 41$ puncta), and basal large puncta ($n = 48$ puncta). \*\*\*$P = 3.75 \times 10^{-4}$, stimulated bulk cytosol versus stimulated puncta; \*\*\*\*$P < 1 \times 10^{-15}$, stimulated bulk cytosol versus basal large puncta; and \*\*\*\*$P = 2.07 \times 10^{-11}$, stimulated puncta versus basal large puncta; Brown-Forsythe and Welch one-way ANOVA with Dunnett's T3 multiple comparisons test. Error bars indicate mean ± SEM. All scale bars: 10 μm. The data underlying this figure can be found in S1 Data.

3-isobutyl-1-methylxanthine (IBMX) to elevate cAMP levels. As previously seen in other cell types [8], RIα-GFP2 knockin MIN6 β cells also exhibited the formation of new, small fluorescent puncta in response to cAMP elevation ($n = 16$ cells, $P < 0.0001$) (Fig 1B). Treatment with GLP-1 also triggered significant accumulation of new, small fluorescent puncta ($n = 23$ cells, $P < 0.0001$) (Fig 1C).

Previously, we showed that binding of the PKA catalytic subunit (PKA-C) inhibits RIα phase separation in the canonical PKA holoenzyme, whereas cAMP binding to RIα induces a conformational change that displaces PKA-C and relieves this inhibitory effect, enabling RIα LLPS while also retaining PKA-C [8]. Consistent with these observations, we found that MIN6 β cells overexpressing both RIα-GFP2 and TagBFP2-PKA-Cα produced large basal puncta that showed overlapping GFP and BFP fluorescence signals, indicating colocalization of RIα and PKA-C (Fig 1D). As above, cAMP elevation via Fsk/IBMX treatment led to the formation of new, small fluorescent puncta ($n = 22$ cells and $P = 9.63 \times 10^{-7}$), which were also positive for both RIα-GFP2 and TagBFP2-PKA-Cα, while the large, pre-existing puncta increased in size by ~1.4-fold (−Fsk/IBMX: $1.58 \pm 0.46$ μm$^2$; +Fsk/IBMX: $2.21 \pm 0.48$ μm$^2$, $n = 16$ puncta from 7 cells, $P = 0.008$) (Fig 1D and 1E). Notably, the newly formed, circular puncta appeared to be highly dynamic and were observed to coalesce on the minute time-scale, indicating liquid-like properties (Fig 1D and S1 Movie). To further investigate RIα condensate dynamics, we performed fluorescence recovery after photobleaching (FRAP) experiments, which indicated that fluorescently tagged RIα can rapidly and dynamically exchange between the small, newly formed puncta and the diffuse, cytosolic pool (normalized recovery = $0.64 \pm 0.03$ s, time to half maximum [$t_{1/2}$] = $4.64 \pm 0.57$ s; $n = 41$ puncta). Conversely, the larger, pre-formed puncta showed much slower FRAP recovery kinetics and reduced apparent diffusion versus the cAMP-induced puncta (normalized recovery = $0.30 \pm 0.03$ s, $P = 2.07 \times 10^{-11}$ versus cAMP-induced puncta; $t_{1/2} = 41.6 \pm 5.6$ s, $P = 1.05 \times 10^{-7}$ versus cAMP-induced puncta; $n = 48$ puncta) (Figs 1F, 1G, and S1B–S1D). cAMP elevations thus induce the formation of new, liquid-like RIα condensates, whereas the large, pre-existing puncta appear to represent matured, gel-like condensates. Overall, these data reveal that RIα undergoes LLPS in MIN6 β cells, yielding condensates with diverse material properties, including dynamic, liquid-like puncta formed in response to physiological cAMP stimulation.

## Loss of RIα LLPS impairs cAMP compartmentalization in MIN6 β cells

We previously showed that RIα biomolecular condensates play a key role in driving cAMP compartmentalization [8,10]. RIα condensates dynamically buffer intracellular cAMP elevations, thereby allowing PDEs to effectively sculpt cAMP signals through local degradation [8]. Thus, we asked if RIα LLPS played a similar role in MIN6 β cells. To study this, we established an RIα-null MIN6 β cell line using CRISPR/Cas9 technology. Western blotting confirmed both the complete absence of endogenous RIα expression (S2A Fig) and that RIα deletion did not significantly affect the expression of other PKA subunits (S2B and S2C Fig). Using these RIα-null MIN6 β cells, we first investigated cAMP buffering by testing whether the loss of RIα condensates altered basal free cAMP levels in the general cytosol in MIN6 β cells. To quantify this effect, we transfected WT and RIα-null MIN6 β cells with diffusible ICUE4, a genetically encoded FRET-based cAMP indicator wherein cAMP binding leads to a conformational change that increases the cyan-over-yellow (C/Y) emission ratio. Comparing the resting ICUE4 emission ratios in the cytosol, we found that RIα-null MIN6 β cells showed higher C/Y emission ratios ($R = 0.64 \pm 0.01$, $n = 232$ cells) compared to WT MIN6 β cells ($R = 0.61 \pm 0.01$, $n = 217$ cells, $P = 8.47 \times 10^{-3}$ versus RIα-null cells), whereas no ratio differences were observed in cells expressing ICUE4 (R279E), a negative-control mutant that lacks cAMP binding (WT: $R = 0.561 \pm 0.008$, $n = 108$ cells; KO: $R = 0.549 \pm 0.008$, $n = 103$ cells, $P = 0.287$) (Fig 2A). These results suggest that the absence of RIα leads to increased basal cAMP accumulation in MIN6 β cells.

PLOS Biology

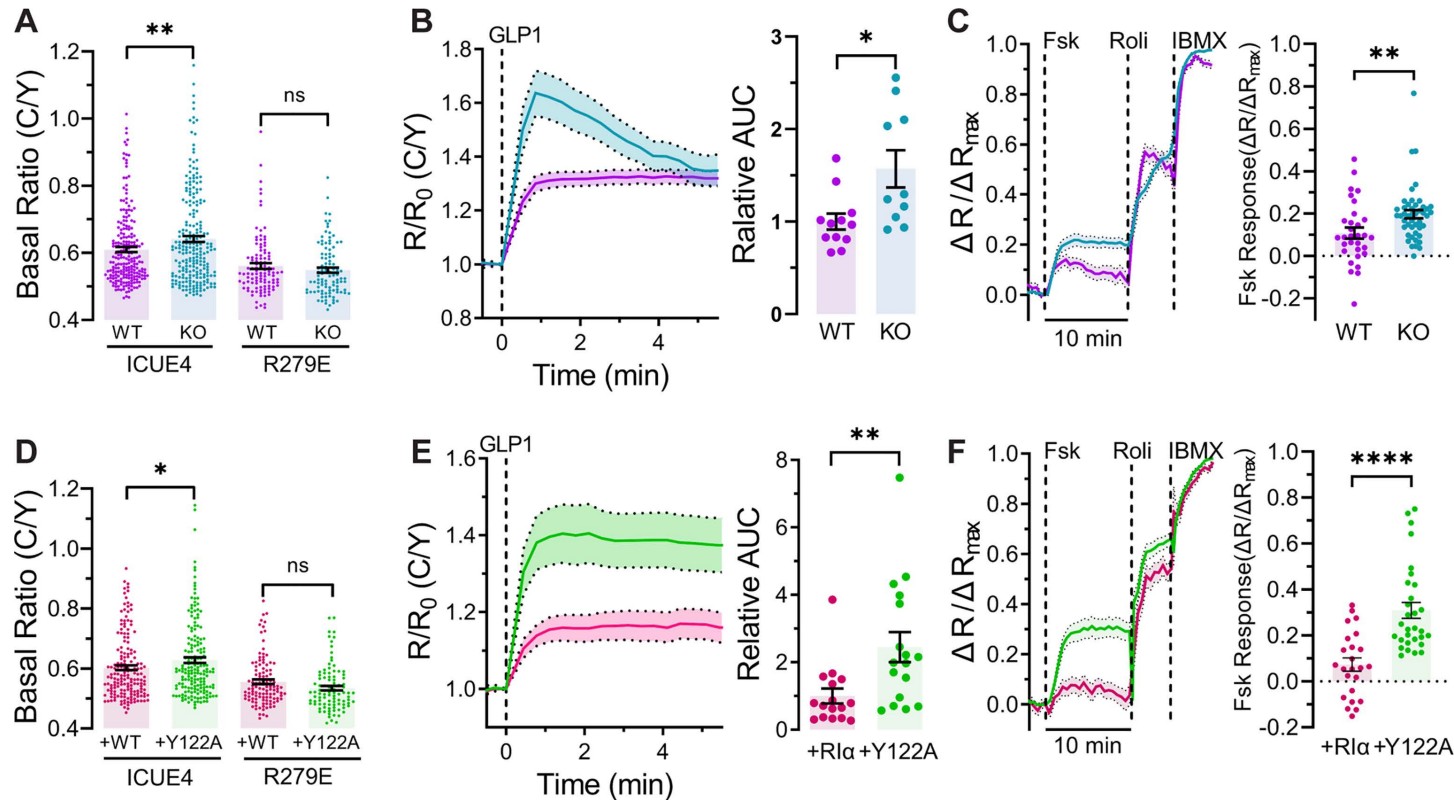

**Fig 2. Loss of RIα LLPS impairs cAMP compartmentalization. (A)** Raw cyan/yellow (C/Y) emission ratios measured in WT and RIα-null MIN6 β cells expressing either ICUE4 or ICUE4 (R279E), showing basal cAMP levels. $n = 217$ (WT) and 232 cells (KO) for ICUE4, and $n = 108$ (WT) and 103 cells (KO) for ICUE4 (R279E). **$P = 8.47 \times 10^{-3}$; ns, $P = 0.29$; unpaired, two-tailed Student $t$ test. Error bars indicate mean ± SEM. **(B)** Time course of normalized C/Y emission ratios in WT and RIα KO MIN6 β cells expressing cAMPFIRE, showing cytosolic cAMP levels **(left)**. Cells were stimulated with GLP-1 (1 μM) at $t_0$. Area under curve (AUC) analysis from $t_0$ to $t_{5min}$, normalized to wild-type values **(right)**. $n = 12$ (WT) and $n = 10$ (KO). *$P = 0.0116$, unpaired two-tailed Student $t$ test. Data represent mean ± SEM. **(C)** Time course of the max-normalized emission ratio change ($\Delta R/\Delta R_{max}$) **(left)** and quantification of Fsk-stimulated response **(right)** in WT ($n = 31$ cells) and RIα KO ($n = 46$ cells) MIN6 β cells expressing PDE4D2$_{cat}$-ICUE4 and sequentially stimulated with Fsk (50 μM), Roli (1 μM), IBMX (100 μM). Solid lines indicate the mean, and shaded areas indicate SEM. **P = 0.0065; unpaired, two-tailed Student $t$ test. **(D)** Raw C/Y emission ratios measured in RIα-null MIN6 β cells expressing either RIα$^{WT}$ (WT) or RIα$^{Y122A}$ (Y122A) plus either ICUE4 or ICUE4 (R279E). $n = 113$ (WT) and 96 cells (Y122A) for ICUE4, and $n = 170$ (WT) and 177 cells (Y122A) for ICUE4 (R279E). *$P = 0.043$; ns, $P = 0.057$; unpaired, two-tailed Student $t$ test. Error bars indicate mean ± SEM. **(E)** Time course of normalized C/Y emission ratios of cAMPFIRE in RIα-null MIN6 β cells expressing either RIα$^{WT}$ (WT) or RIα$^{Y122A}$ (Y122A). Cells were stimulated with GLP-1 (1 μM) at $t_0$ **(left)**. Area under curve (AUC) analysis from $t_0$ to $t_{5min}$, normalized to RIα$^{WT}$ **(right)**. $n = 16$ (RIα$^{WT}$) and $n = 17$ (RIα$^{Y122A}$). **$P = 0.0079$, unpaired two-tailed Student $t$ test. Data represent mean ± SEM. **(F)** Time course of the max-normalized emission ratio change ($\Delta R/\Delta R_{max}$) **(left)** and quantification of Fsk-stimulated response **(right)** in RIα-null MIN6 β cells expressing either RIα$^{WT}$ (WT, $n = 24$ cells) or RIα$^{Y122A}$ (Y122A, $n = 30$ cells) plus PDE4D2$_{cat}$-ICUE4 and stimulated as in (B). Solid lines indicate the mean, and shaded areas SEM. ****$P = 5.17 \times 10^{-6}$; unpaired, two-tailed Student $t$ test. The data underlying this figure can be found in S1 Data.

Given that cAMP buffering also likely plays a role in modulating receptor-stimulated cAMP elevations [11], we next investigated whether loss of RIα would promote higher cAMP accumulation upon GLP-1 stimulation. Indeed, using the higher-sensitivity cAMP indicator cAMPFIRE [12], we observed significantly greater cAMP-induced responses in RIα-null MIN6 β cells compared to WT cells following GLP-1 stimulation (WT: $1.000 \pm 0.0859$, $n = 12$; KO: $1.571 \pm 0.2009$, $n = 10$, $P = 0.0116$) (Fig 2B). These results suggest that the loss of RIα expression in MIN6 β cells enhances both basal and stimulated cAMP accumulation in the cytosol compared to WT MIN6 β cells, consistent with a loss of cAMP buffering.

Given these alterations in cytosolic cAMP dynamics, we next sought to quantitatively assess the impact of RIα on cAMP compartmentalization, using our previously developed cAMP compartmentalization assay [8]. A key driver of cAMP

compartmentalization is the ability of PDEs degrade cAMP in their immediate vicinity, creating local cAMP sinks that act as diffusional barriers to separate distinct cAMP microdomains [6]. Our assay utilizes ICUE4 C-terminally tethered to the catalytic subunit of PDE4D2 (PDE4D2$_{cat}$) to directly probe this phenomenon. Specifically, if compartmentalization is intact, local cAMP degradation by PDE4D2$_{cat}$ will suppress the response from tethered ICUE4, whereas loss of compartmentalization, such as through disrupted RIα LLPS, will rescue the ICUE4 response (S2D Fig). Wildtype (WT) or RIα-null MIN6 β cells expressing PDE4D2$_{cat}$-ICUE4 were first stimulated with Fsk to induce cAMP production, followed by the PDE4-specific inhibitor Rolipram to assess the impact of PDE4 activity on the Fsk-induced sensor response, and finally, the pan-PDE inhibitor IBMX to maximally elevate cAMP and calibrate the sensor response. Consistent with our hypothesis, we observed that Fsk stimulation triggered a 1.82-fold higher max-normalized C/Y emission ratio change ($\Delta R/\Delta R_{max}$) in RIα-null MIN6 β cells ($\Delta R/\Delta R_{max} = 0.197 \pm 0.019$, $n = 46$ cells) compared with WT MIN6 β cells ($\Delta R/\Delta R_{max} = 0.108 \pm 0.026$, $n = 31$ cells, $P = 0.0065$), suggesting that knocking out RIα in MIN6 β cells attenuates cAMP compartmentalization by PDE4D2 (Fig 2C).

Although disruption of PDE-mediated cAMP compartmentalization and elevation of cytosolic cAMP levels are both consistent with the loss of cAMP buffering by RIα condensates, knocking out RIα can unleash active PKA-C, which has been shown to phosphorylate both ACs and PDEs [13,14], potentially influencing cAMP levels via feedback mechanisms. We therefore sought a more selective means to perturb RIα LLPS without affecting the PKA holoenzyme. Our recent study identified an RIα mutant with reduced ability to undergo LLPS. This mutant, RIα$^{Y122A}$, has a weakened N3A dimerization interface in the RIα CNB-A domain and exhibits defective cAMP-stimulated LLPS. Indeed, in HEK293T cells expressing fluorescently tagged RIα$^{Y122A}$, Fsk/IBMX stimulation failed to induce a significant change in RIα puncta number (S3A Fig). At the same time, the Y122A mutation does not alter cAMP binding [9], AKAP recruitment (S3B Fig), or PKA-C dissociation (S3C Fig), suggesting an otherwise intact PKA holoenzyme. Critical saturation concentration (c-sat) curves derived from in vitro droplet formation assays reveal that RIα$^{Y122A}$ exhibits more restricted LLPS behavior compared to RIα$^{WT}$, being strongly inhibited by PKA-C (S3D Fig). When mRuby2-tagged RIα$^{Y122A}$ was introduced into RIα-null MIN6 β cells, fewer puncta were observed at the basal state compared to RIα$^{WT}$ (RIα$^{WT}$: median of 6 puncta per cell, $n = 98$ cells, 95% confidence interval (CI): 5–7; RIα$^{Y122A}$: median of 2 puncta per cell, $n = 103$ cells, 95% CI: 2–3; $P = 5.47 \times 10^{-12}$) (S3E Fig). Consistent with our observations in HEK293T cells, RIα-null MIN6 β cells expressing GFP2-tagged RIα$^{Y122A}$ did not show any significant changes in puncta number or size upon Fsk/IBMX stimulation, indicating a loss of cAMP-induced LLPS (S3F Fig and S2 Movie). We therefore reasoned that the RIα$^{Y122A}$ mutant can be used as a molecular tool to more precisely test the specific functional impact of abolishing RIα LLPS.

First, we examined cytosolic cAMP levels using diffusible ICUE4 in RIα-null MIN6 β cells expressing either RIα$^{WT}$ or RIα$^{Y122A}$. We found that RIα$^{Y122A}$-expressing RIα-null MIN6 β cells exhibited higher basal ICUE4 emission ratios ($R = 0.628 \pm 0.00955$, $n = 177$ cells) than cells expressing RIα$^{WT}$ ($R = 0.603 \pm 0.00819$, $n = 170$ cells, $P = 0.0434$), indicating that basal cytosolic cAMP levels rise when RIα LLPS is disrupted (Fig 2D). RIα$^{Y122A}$-expressing RIα-null MIN6 β cells also exhibited a significantly higher cAMP response to GLP-1 stimulation (RIα$^{WT}$: $1.000 \pm 0.2241$, $n = 16$; RIα$^{Y122A}$: $2.449 \pm 0.4471$, $n = 17$, $P = 0.0079$) (Fig 2E). Furthermore, when we repeated our cAMP compartmentalization assay in RIα-null MIN6 β cells co-transfected with either mRuby2-tagged RIα$^{WT}$ or RIα$^{Y122A}$ plus PDE4D2$_{cat}$-ICUE4, we found that RIα$^{Y122A}$-expressing cells showed an approximately 4.23-fold higher max-normalized C/Y emission ratio change in response to Fsk stimulation ($\Delta R/\Delta R_{max} = 0.309 \pm 0.034$, $n = 30$ cells) than RIα$^{WT}$-expressing cells ($\Delta R/\Delta R_{max} = 0.073 \pm 0.029$, $n = 24$ cells, $P = 5.17 \times 10^{-6}$), indicating that cAMP compartmentalization was compromised by the selective loss of cAMP-dependent RIα LLPS (Fig 2F). Together, our results show that RIα condensates enable effective cAMP compartmentalization and critically regulate cAMP levels in MIN6 β cells.

## RIα LLPS regulates MIN6 β cell function

Considering the contribution of RIα LLPS to cAMP compartmentalization and the importance of specific cAMP signaling in regulating numerous cellular functions [15], we hypothesized that RIα LLPS plays an important role in MIN6 β cell

function. In β cells, PDEs are critical regulators of the oscillatory signaling circuit, where they modulate the frequency of both cAMP and $Ca^{2+}$ oscillations through local degradation of cAMP [5]. These oscillatory patterns are particularly crucial as they directly influence insulin secretion, a primary function of β cells [4]. Given this intricate relationship between cAMP compartmentalization and β cell function, we first investigated whether disruption of RIα LLPS alters $Ca^{2+}$ and cAMP oscillations, two critical axes of this oscillatory signaling circuit. We visualized $Ca^{2+}$ dynamics using the genetically encoded $Ca^{2+}$ indicator RCaMP [16] in RIα-null MIN6 β cells co-expressing either RIα$^{WT}$ or RIα$^{Y122A}$ and stimulated with the $K^+$ channel blocker tetraethylammonium chloride (TEA). Interestingly, when we measured the peak-to-peak interval between $Ca^{2+}$ spikes, we found that RIα$^{Y122A}$-expressing RIα-null MIN6 cells showed approximately 1.29-fold faster $Ca^{2+}$ oscillations (134.1 ± 9.64 s, $n = 37$ cells) than cells expressing RIα$^{WT}$ (172.9 ± 10.91 s, $n = 34$ cells, $P = 9.2 \times 10^{-3}$) (Fig 3A). We

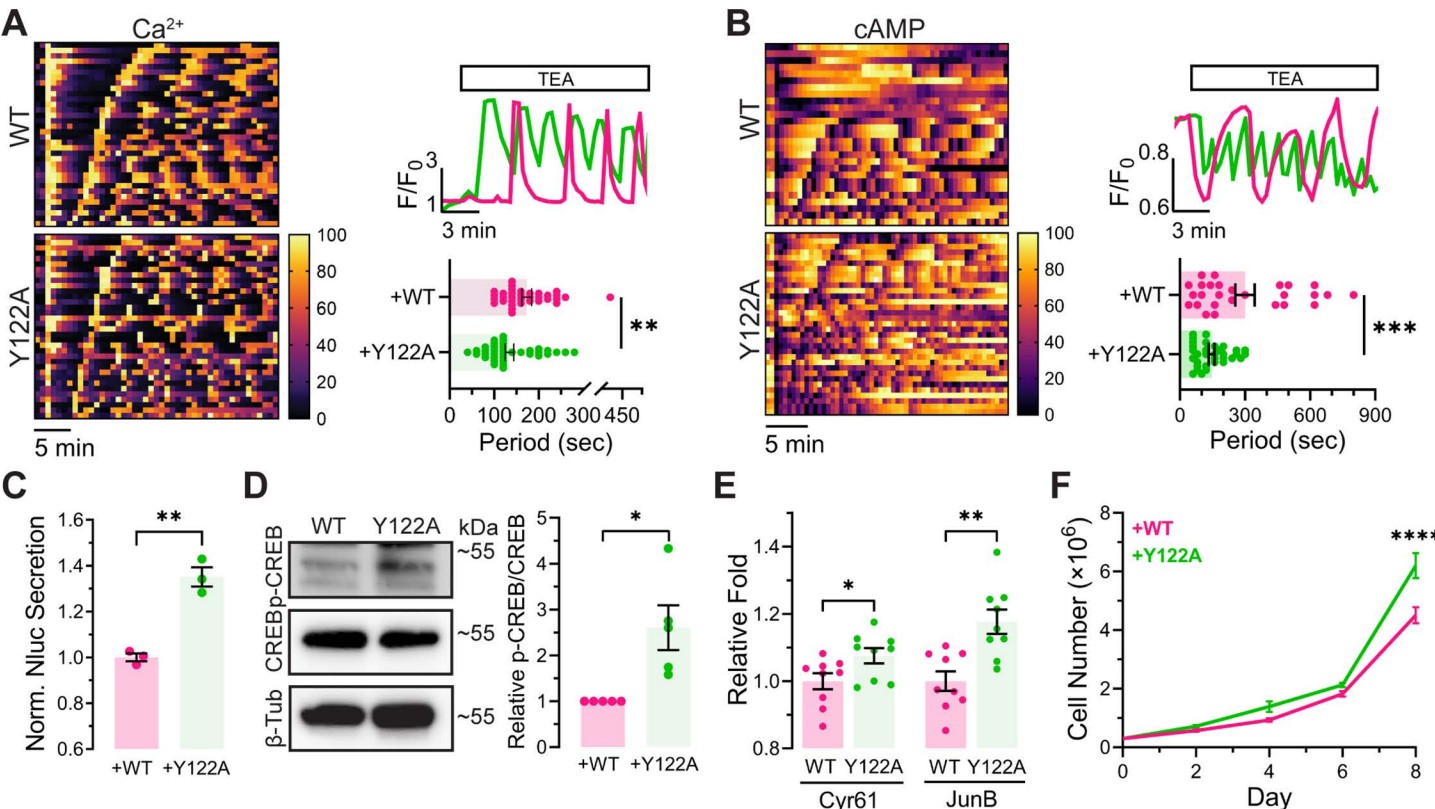

**Fig 3. RIα LLPS regulates MIN6 β cell function. (A)** Heatmap **(left)** showing $Ca^{2+}$ oscillations in individual RIα-null MIN6 β cells (each row) expressing either RIα$^{WT}$ (WT, upper) or RIα$^{Y122A}$ (Y122A, lower), plotted as normalized RCaMP intensity. Cells were ordered by the period of the first two $Ca^{2+}$ peaks after TEA stimulation. Representative single-cell traces of $Ca^{2+}$ oscillation **(top right)** and quantification of oscillation period **(bottom right)** in WT ($n = 34$ cells) or Y122A ($n = 37$ cells). **$P = 9.2 \times 10^{-3}$; unpaired, two-tailed Student $t$ test. Data represent mean ± SEM. **(B)** Heatmap **(left)** depicting temporal dynamics of cAMP oscillations in individual RIα-null MIN6 β cells (each row) expressing either RIα$^{WT}$ (WT, upper) or RIα$^{Y122A}$ (Y122A, lower), shown as normalized cAMPFIRE C/Y emission ratios. Cell ordering reflects the period between the initial two cAMP peaks following TEA stimulation. Representative cAMP oscillation traces **(top right)** and period quantification **(bottom right)** in WT ($n = 33$ cells) or Y122A ($n = 27$ cells). ***$P = 0.0006$; unpaired, two-tailed Student $t$ test. Data represent mean ± SEM. **(C)** Relative insulin secretion level between RIα-null MIN6-6 cells expressing RIα$^{WT}$ (WT) or RIα$^{Y122A}$ (Y122A) upon glucose (25 mM)/ GLP-1(1 μM) stimulation. $n = 3$ for both, **$P = 0.0016$; unpaired, two-tailed Student $t$ test. Error bars indicate mean ± SEM. **(D)** Western blot of CREB and phospho-CREB (p-CREB) **(top)** and quantification of p-CREB/CREB **(bottom)** in RIα-null MIN6 β cells expressing RIα$^{WT}$ (WT) or RIα$^{Y122A}$ (Y122A). *$P = 0.011$; unpaired, two-tailed Student $t$ test ($n = 5$ experiments). **(E)** Quantification of Cyr61 and JunB mRNA levels in RIα-null MIN6 β cells expressing RIα$^{WT}$ (WT) or RIα$^{Y122A}$ (Y122A). *$P = 0.036$; **$P = 0.0015$; unpaired, two-tailed Student $t$ test. **(F)** Quantification of cell proliferation in RIα-null MIN6 β cells expressing RIα$^{WT}$ (WT) or RIα$^{Y122A}$ (Y122A) ($n = 4$ experiments). ****$P = 6.98 \times 10^{-5}$; Multiple paired $t$ test. Error bars indicate mean ± SEM. The data underlying this figure can be found in S1 Data and S1 Raw Images.

observed similar results when we compared RIα-null and WT MIN6 β cells, with knockout cells exhibiting approximately 1.60-fold faster TEA-stimulated $Ca^{2+}$ oscillations than WT ($160 \pm 12.75$ s, $n = 61$ cells versus $255.47 \pm 24.21$ s, $n = 53$ cells, $P = 4 \times 10^{-4}$) (S4A Fig). We next monitored cAMP oscillations under the same conditions using cAMPFIRE. Consistent with the observed alteration in $Ca^{2+}$ dynamics, TEA-induced cAMP oscillations were 2.05-fold faster in $RIα^{Y122A}$-expressing RIα-null cells versus $RIα^{WT}$-expressing cells ($RIα^{WT}$: $300 \pm 44.11$ s, $n = 27$; $RIα^{Y122A}$: $146.1 \pm 13.38$ s, $n = 33$, $P = 0.0006$) (Fig 3B) and 1.92-fold faster in RIα-null cells than in WT MIN6 β cells (WT: $197.9 \pm 26.44$ s, $n = 19$; RIα-null: $103.2 \pm 8.901$ s, $n = 31$, $P = 0.0002$) (S4B Fig). These data suggest that cAMP compartmentalization enabled by RIα LLPS is important for controlling $Ca^{2+}$ and cAMP oscillation frequency, consistent with previous work demonstrating that PKA serves as a frequency modulator in this oscillatory circuit [5].

Having established that RIα LLPS regulates the frequency of both $Ca^{2+}$ and cAMP oscillations, we next investigated whether these alterations in the oscillatory circuit affect insulin secretion, given the established role of cAMP signaling in this process [4]. To this end, we utilized an engineered MIN6 β cell line (MIN6-6) expressing a modified preproinsulin construct where NanoLuc luciferase (NLuc) is inserted into the C-peptide region [17]. This system allowed us to assess insulin secretion under glucose and/or GLP-1-stimulation conditions (S5A Fig). We first established an RIα-null MIN6-6 cell line (S5B Fig) to test whether RIα is involved in insulin secretion. RIα-null MIN6-6 cells demonstrated significantly reduced insulin secretion compared to parent cells when stimulated with glucose/GLP-1 ($P = 0.0001$ and $n = 3$) (S5C Fig), indicating that RIα plays a significant role in insulin secretion. We then compared insulin secretion in RIα-null MIN6-6 cells expressing either $RIα^{Y122A}$ or $RIα^{WT}$. Interestingly, cells expressing $RIα^{Y122A}$ exhibited enhanced insulin secretion ($P = 0.0016$, $n = 3$) (Fig 3C), consistent with the elevated cAMP levels and faster $Ca^{2+}$ and cAMP oscillations. In contrast, a mutant that has the same cAMP binding characteristics as $RIα^{Y122A}$ but retains cAMP-dependent LLPS, $RIα^{K123A}$ [9], led to reduced insulin secretion compared to cells expressing $RIα^{Y122A}$ ($P = 0.0078$ and $n = 3$) (S5D and S5E Fig). These findings suggest that RIα LLPS suppresses insulin secretion.

Beyond its local effects on oscillatory signaling and insulin secretion, we explored whether RIα LLPS could also influence global cAMP signaling. cAMP-response element binding protein (CREB), a key mediator of cAMP-dependent transcription, is activated via phosphorylation at S133 by a variety of protein kinases, including PKA, to initiate transcription of target genes [18]. Given the global elevation of cAMP levels observed in $RIα^{Y122A}$-expressing MIN6 β cells (Fig 2D), we hypothesized that these cells would also show elevated CREB phosphorylation. We therefore probed phospho-CREB levels in RIα-null MIN6 β cells expressing $RIα^{WT}$ or $RIα^{Y122A}$ by immunoblotting with a phospho-CREB-specific antibody. Indeed, phospho-CREB levels were significantly higher in $RIα^{Y122A}$-expressing RIα-null MIN6 β cells versus $RIα^{WT}$-expressing cells ($2.604 \pm 0.49$-fold, $n = 5$, $P = 0.0113$) (Fig 3D). Similarly, elevated phospho-CREB levels were also detected in RIα-null MIN6 β cells compared to WT MIN6 β cells ($n = 4$ experiments, $P = 2.62 \times 10^{-3}$) (S6A Fig). These observations suggest that RIα LLPS plays a role in regulating basal CREB phosphorylation in MIN6 β cells. Given these findings, we hypothesized that RIα LLPS would also impact the transcription of CREB target genes in MIN6 β cells. We therefore analyzed representative CREB target genes involved in cell proliferation, Cyr61 [19,20] and JunB [21,22], through reverse transcription PCR (RT-PCR). RIα-null MIN6 β cells expressing $RIα^{Y122A}$ showed increased expression of both Cyr61 and JunB versus $RIα^{WT}$-expressing RIα-null cells (Cyr61: $1.076 \pm 0.023$-fold, $n = 9$, $P = 0.0363$; JunB: $1.177 \pm 0.036$-fold, $n = 9$, $P = 0.0015$) (Fig 3E), as did RIα-null MIN6 β when compared with WT MIN6 β cells (Cyr61: $2.08 \pm 0.074$-fold, $P = 5.9 \times 10^{-8}$ and JunB: $1.79 \pm 0.123$-fold for, $P = 6.68 \times 10^{-4}$, $n = 7$ experiments) (S6B Fig). These results indicate increased CREB target gene transcription resulting from the disruption of RIα LLPS.

Based on our findings that disrupting RIα LLPS leads to upregulation of proliferation-related CREB targets (Figs 3E and S6B), as well as previous observations that RIα knockout increases cell proliferation in non-tumorigenic AML12 hepatocytes [8], we further examined whether the loss of RIα LLPS could induce increased proliferation in MIN6 β cells. Indeed, we found that RIα-null MIN6 β cells expressing $RIα^{Y122A}$ showed 1.37-fold greater proliferation ($n = 4$ experiments, $P = 6.98 \times 10^{-5}$) than RIα-null cells expressing $RIα^{WT}$ (Fig 3F). Similarly, RIα-null MIN6 β cells alone showed a 2.04-fold

increase in proliferation compared with WT MIN6 β cells ($n = 4$ experiments, $P = 1.82 \times 10^{-3}$) (S6C Fig), revealing that maintenance of proper cAMP levels through RIα LLPS is critical for controlling proliferation of MIN6 β cells. These results collectively highlight the functional significance of RIα LLPS in regulating MIN6 β cell physiology.

## Discussion

Our study demonstrates that RIα undergoes LLPS in MIN6 β cells, which join a growing number of cell types that contain RIα condensates, suggesting LLPS is a conserved property of this ubiquitous protein. In addition to cAMP-stimulated formation of small RIα puncta, we observed the presence of large, pre-formed RIα condensates in MIN6 β cells that are not found in HEK293T cells [8] (Fig 1A and 1D). These pre-formed condensates exhibit slow recovery in FRAP experiments, in contrast to the more rapid recovery dynamics of RIα condensates formed in response to cAMP elevation (Figs 1F, 1G, and S1B–S1D). Condensates with initially liquid-like properties often become increasingly gel-like over time and lose the ability to exchange molecules with the environment, a process called maturation [23]. Given the large excess of RIα in comparison with the catalytic subunit [24], it is possible that RIα basally forms condensates that gradually mature into gel-like structures, whereas newly formed, cAMP-induced RIα condensates remain more liquid-like and reversible. All of these structures could serve as cAMP reservoirs in MIN6 β cells, potentially helping buffer cytosolic cAMP levels to achieve precise spatial regulation.

It is increasingly clear that LLPS of biomolecules is critically involved in regulating cellular processes and function [8,25–29]. However, pinpointing the functional roles of specific biomolecular condensates is often challenging. Part of the difficulty lies in isolating the effect of LLPS from other properties of the target protein. The typical methods to control LLPS include manipulating intrinsically disordered regions within driver proteins via insertion or deletion or treating cells with 1,6-hexanediol, which interferes with the weak intermolecular interactions that underlie LLPS. However, these interactions are also important for other molecular processes, leading 1,6-hexanediol to exhibit nonspecific effects [10,30]. A more ideal strategy for studying the functional consequences of LLPS would be to develop tools that selectively disrupt LLPS by the target protein without affecting other functions. Here, we were able to selectively control RIα LLPS through the introduction of a single Y122A substitution, which disrupts a secondary dimerization site involved in driving phase separation. Previously, the RIα[Y122A] mutant was shown to exhibit defective cAMP-induced LLPS in HEK293T cells [10], and we observed a similar effect in MIN6 β cells (S3F Fig). Consistent with previous observations, however, neither cAMP-induced dissociation of the RIα dimer from PKA-C nor recruitment of the RIα dimer to A-kinase anchoring proteins (AKAPs), two critical signaling functions that are also disrupted by LLPS-blocking mutations in the RIα cyclic-nucleotide-binding domains (E202A/E326A) or docking/dimerization domain (L50R) [10], respectively, was affected by the Y122A mutation (S3B and S3C Fig). Thus, although multiple mutations have been shown to interfere with RIα LLPS, only the Y122A mutation does not affect other integral functions of the PKA holoenzyme, making this mutant a useful tool to dissect the functional roles of RIα LLPS. Indeed, while cells lacking RIα or re-expressing RIα[Y122A] yielded largely similar results, we observed strikingly opposite effects on insulin secretion, presumably because RIα plays multiple roles in this process, thus underscoring the importance of using molecular tools that specifically perturb condensates.

By utilizing the Y122A mutation to selectively disrupt RIα LLPS, we were able to directly connect RIα phase separation behavior to the regulation of MIN6 β cell functions. In particular, our study demonstrates a key role for RIα condensates in regulating β-cell function through cAMP signaling compartmentalization. We observed a striking loss of PDE-mediated cAMP compartmentalization in RIα-null cells expressing RIα[Y122A], resulting in increased cAMP levels, both locally within PDE microdomains and globally in the cytosol (Fig 2). In β cells, cAMP and $Ca^{2+}$ have been shown to undergo synchronized oscillations [31,32], and we previously discovered that cAMP, PKA, and $Ca^{2+}$ in fact form a highly integrated, oscillatory signaling circuit that controls various cellular processes [5]. Here, we found that elevated cAMP levels caused by abolishing RIα LLPS increased the frequency of both $Ca^{2+}$ and cAMP oscillations in TEA-stimulated MIN6 β cells

(Figs 3A, 3B, S4A, and S4B), which mirrors our previous observation of more rapid $Ca^{2+}$ oscillations following inhibition of PDE activity [5]. The temporal dynamics of cAMP and $Ca^{2+}$ oscillations are highly compartmentalized, exhibiting distinct phase relationships within $Ca^{2+}$ channel nanodomains versus the general plasma membrane [7]. Recruitment of a specific AC isoform to these nanodomains via a specific AKAP plays an important role in this local regulation, disruption of which leads to defective $Ca^{2+}$ oscillations [7]. Maintaining low basal cAMP levels would seem critical for such precise cAMP compartmentalization, and RIα condensates may help sculpt these signaling nanodomains. Future studies will thus investigate the role of RIα LLPS in nanoscale cAMP compartmentalization, be it within proximity of $Ca^{2+}$ channels, as discussed in a previous study [7], or around GLP-1 receptors as recently demonstrated [33].

Similar to previous results implicating loss of cAMP compartmentalization and aberrant signaling in increased cell proliferation [8], RIα-null MIN6 β cells expressing RIα^Y122A proliferated more rapidly than RIα-null cells expressing RIα^WT (Fig 3F). The observed increase in cell proliferation may be due to elevated CREB phosphorylation and subsequent induction of proliferation-associated CREB target genes, such as Cyr61 and JunB [19–22], in these cells (Figs 3D, 3E, S6A, and S6B). The increase in CREB phosphorylation can be attributed to elevated cytosolic cAMP levels in RIα-null MIN6 β cells expressing RIα^Y122A. This enhanced CREB phosphorylation might also reflect the distinct spatial organization of PKA signaling. While RII is known to be locally targeted to specific membrane compartments, for example, through AKAP79 to regulate $Ca^{2+}$ channels and $Ca^{2+}$ oscillations [5,7,34], RI may have broader cellular effects. In addition, the increased $Ca^{2+}$ and cAMP oscillation frequency observed in these cells could also help switch PKA signaling to a global activation mode, leading to increased CREB phosphorylation [5]. Future studies aimed at exploring the broader impact of RIα LLPS on gene expression changes, such as through RNAseq, will be important to provide a more comprehensive picture of how RIα LLPS regulates MIN6 β cell function. Moreover, our findings reveal that RIα LLPS functions to suppress insulin secretion. Based on previous reports that PKA activity enhances acute insulin secretion [35], we hypothesize that RIα-mediated LLPS may serve as a regulatory mechanism for acute insulin secretion in β cells through its demonstrated role in cAMP buffering, a compelling avenue for future investigation.

Using the Y122A mutant to probe RIα LLPS is not without limitations. For example, RIα^Y122A has been shown to exhibit increased cAMP sensitivity and reduced cooperativity [9], which may subtly affect cAMP regulation of PKA signaling. Nevertheless, RIα^Y122A has greatly facilitated our study and advanced our understanding of the functional roles of RIα LLPS. Given the ubiquitous expression of RIα, which is especially abundant in central nervous system and cardiac tissues [36], we expect RIα^Y122A to serve as an important tool to study RIα LLPS and uncover the functional landscape of this ubiquitous biomolecular condensate.

## Materials and methods

| REAGENT or RESOURCE | SOURCE | IDENTIFIER |
|---|---|---|
| Bacterial and Virus Strains | | |
| DH5α competent bacteria | NEB | Cat# C2987I |
| Experimental Models: Cell Lines | | |
| MIN6 | Miyazaki Lab | N/A |
| HEK293T | ATCC | Cat# CRL-11268 |
| 293T-RIα KO | Zhang and colleagues [8] | N/A |
| MIN6-6 | Chen Lab | N/A |
| Chemicals, Peptides, and Recombinant Proteins | | |
| Q5 High Fidelity Polymerase | NEB | Cat# M0491S |
| HiFi DNA Assembly Kit | NEB | Cat# E5520S |
| SuperScript III One-step RT-PCR Kit | Invitrogen | Cat# 12574-018 |
| PfuUltra II Fusion HS DNA Polymerase | Agilent | Cat# 600670-51 |
| TEA | Sigma | Cat# T2265 |

PLOS Biology

| REAGENT or RESOURCE | SOURCE | IDENTIFIER |
|---|---|---|
| Forskolin | CalBioChem | Cat# 344281 |
| IBMX | Sigma Aldrich | Cat# I7018 |
| Rolipram | Alexis | Cat# 61413-54-5 |
| Isoproterenol | Sigma Aldrich | Cat# 1351005 |
| CTZ400a (DeepBlueC) | Nanolight Technology | Cat# 340 |
| DMEM:F12 | ThermoFisher | Cat# 12634010 |
| Fetal bovine serum (FBS) | GIBCO | Cat# 26140079 |
| Penicillin-streptomycin (Pen-Strep) | GIBCO | Cat# 15140122 |
| β-mercaptoethanol | Sigma | Cat# M3148 |
| PolyJet | Signagen | Cat# SL100688 |
| Lipofectamine 2000 | Invitrogen | Cat# 11668-019 |
| Poly-D-Lysine | Sigma Aldrich | Cat# P6407 |
| Hank's balanced salt solution (HBSS) | GIBCO | Cat# 14065 |
| Protease Inhibitor | Roche Applied Science | Cat# 11873580001 |
| SuperSignal West Femto | Thermo Scientific | Cat# 34094 |
| Antibodies | | |
| Rabbit anti-β-tubulin antibody | Cell Signaling Technology | Cat# 2146S |
| Rabbit anti-GAPDH | Cell Signaling Technology | Cat# 2118S |
| Mouse anti-RIα antibody | BD Biosciences | Cat# 610610 |
| Sheep anti-RIβ antibody | R&D Systems | Cat# AF1477 |
| Mouse anti-RIIα antibody | BD Biosciences | Cat# 612242 |
| Mouse anti-RIIβ antibody | BD Biosciences | Cat# 612550 |
| Mouse anti-PKA[C] antibody | BD Biosciences | Cat# 610980 |
| Rabbit anti-PKA C-α antibody | Cell Signaling Technology | Cat# 4782S |
| Rabbit anti-CREB antibody | Cell Signaling Technology | Cat# 9197S |
| Rabbit anti-phospho-CREB antibody | Cell Signaling Technology | Cat# 9198S |
| Goat anti-mouse IgG-HRP | Thermo Scientific | Cat# PI31430 |
| Goat anti-rabbit IgG-HRP | Thermo Scientific | Cat# PI31460 |
| Rabbit anti-sheep IgG-HRP | Invitrogen | Cat# 31480 |
| Goat anti-rabbit IgG-Alex647 | Invitrogen | Cat# A21245 |
| Oligonucleotides | | |
| Primers for plasmid construction, see Table 2.1 | This paper | N/A |
| Recombinant DNA | | |
| pcDNA3.1 GFP2-RIα | Day and colleagues [37] | N/A |
| pcDNA3.1 GFP2-RIα$^{Y122A}$ | Hardy and colleagues [10] | N/A |
| pcDNA3.1 mCherry-PKA-Ca | Day and colleagues [37] | N/A |
| pcDNA3.1 mRuby2-RIα | Zhang and colleagues [8] | N/A |
| pcDNA3.1 mRuby2-RIα$^{Y122A}$ | This paper | N/A |
| pcDNA3.1 GFP2-C3-PKA_hCa | Hardy and colleagues [10] | N/A |
| pRluc8-N3-hRIα | Isensee and colleagues [38] | N/A |
| pRluc8-N3-hRIα$^{Y122A}$ | Hardy and colleagues [10] | N/A |
| pEGFP-N1 smAKAP-FLAG-EGFP | Hardy and colleagues [10] | N/A |
| RCaMP | Akerboom and colleagues [16] | N/A |
| ICUE4 | Addgene | Cat# 181846 |
| ICUE4$^{R279E}$ | Zhang and colleagues [8] | N/A |
| PDE4D2$_{cat}$-ICUE4 | Zhang and colleagues [8] | N/A |

| REAGENT or RESOURCE | SOURCE | IDENTIFIER |
|---|---|---|
| pCAGGS-cAMPFIRE-M | Zhong lab [12] | N/A |
| Software and Algorithms | | |
| MATLAB (R2019a) | MathWorks | https://www.mathworks.com/products/matlab.html |
| PRISM (10.1.0) | Graphpad | https://www.graphpad.com/scientific-software/prism/ |
| Adobe Illustrator (26.5) | Adobe | https://www.adobe.com/products/illustrator.html |
| Fiji (ImageJ) | NIH | https://imagej.net/software/fiji/downloads |
| METAFLUOR (7.7) | Molecular Devices | https://www.moleculardevices.com/products/cellular-imaging-systems/acquisition-and-analysis-software/metamorph-microscopy |

## Experimental model and subject details

**Cell culture and transfection.** MIN6, MIN6-6, HEK293T, and HEK293T RIα-null cells were maintained in a 5% $CO_2$-controlled humid incubator at 37 °C. MIN6 and MIN6-6 cells were cultured in DMEM (10% fetal bovine serum (FBS), 75 µM β-mercaptoethanol, 10% FBS, 4.5 g/L glucose, and 1% penicillin/streptomycin). HEK293T and RIα-null HEK293T cells were cultured under the same conditions as MIN6 β cells, except that β-mercaptoethanol was excluded from the culture medium. For transfection, cells were grown to ~70% confluency in 35 mm glass-bottom dishes, after which 500 ng - 1.5 µg of plasmid DNA was introduced into MIN6 β cells using Lipofectamine 2000 (Invitrogen) or into HEK293T cells using PolyJet (Signagen). Transfected cells were cultured for 24–48 h before imaging.

**Generation of RIα-null MIN6 β cells.** The pair of guide RNAs targeting PKA RIα was designed using the Synthego Knockout Guide Design (https://design.synthego.com/#/) (S1 Table). The guide RNA pair was cloned into the px459 V2 vector via Golden Gate Assembly using BbsI digestion. Introduction of the vector into MIN6 β cells resulted in the induction of a frame-shift mutation within the PKA RIα genomic locus via non-homologous end joining. Transfected cells were selected on 2 µg/ml puromycin for 3 days starting 24 h after transfection. Loss of RIα protein expression was confirmed by immunoblotting.

**Generation of RIα-GFP2 knock-in MIN6 β cells.** The guide RNA pair targeting PKA RIα C-terminus (S1 Table) was cloned into the px459 V2 vector via Golden Gate Assembly using BbsI digestion. For CRISPR knock-in via homology-directed repair (HDR), homology arms near the RIα C-terminal were amplified using MIN6 β cell genomic DNA (S1 Table). To enable GFP2 insertion at the targeting region, homology arms were attached before and after GFP2 using Gibson assembly to create the HDR template. This template was introduced into a CMV promoter-removed pcDNA3 vector and amplified for transfection. The engineered px459 V2 vector and HDR template-containing vector were introduced into MIN6 β cells at a 1:1 ratio using Lipofectamine 2000. After 48 h of incubation, cells were selected with 2 µg/ml puromycin for 3 days. The successful insertion of GFP2 at the RIα C-terminus was confirmed by fluorescence microscopy, gDNA sequencing, and western blot.

## Method details

**pcDNA3.1 mRuby2-RIα$^{Y122A}$ plasmid construction.** The Y122A point mutation was introduced into mRuby2-RIα [8] via PCR amplification using primers listed in S1 Table, followed by Gibson Assembly using the NEBuilder Hi-Fi DNA Assembly Kit (New England Biolabs). The mutant construct was verified by Sanger sequencing (Genewiz).

**Immunoblotting.** MIN6 β cells were rinsed twice with phosphate-buffered saline (PBS) and harvested. The harvested cells were lysed in radioimmunoprecipitation assay (RIPA) buffer (10 mM Tris-Cl, pH 8.0, 0.1% sodium deoxycholate, 0.5 mM EGTA, 1 mM EDTA, 0.1% SDS, 1% Triton X-100, 140 mM NaCl) containing protease inhibitor cocktail (Roche Applied Science) for 15 min on ice. After centrifugation (12,000 rpm, 4 °C for 15 min), the

supernatant was collected and protein quantification performed using a Pierce BCA Protein Assay kit (Thermo Fisher Scientific). Proteins were separated via SDS-PAGE using 4%–20% Mini-PROTEAN TGX Precast Protein Gels (BioRad), then transferred to PVDF membranes (Millipore). Membranes were blocked for 30 min at room temperature using 5% skim milk in 0.1% TBS-Tween20 (TBST) followed by overnight incubation with diluted primary antibodies (beta-tubulin or GAPDH: 1:5000; all other primary antibodies: 1:1000) at 4 °C. The next day, membranes were washed four times for 5 min each with 0.1% TBST and then incubated with secondary antibody (1:5000) for 1 h at room temperature. Signal was detected by incubating membranes with chemiluminescent HRP substrate (SuperSignal West Femto, Thermo Scientific) and imaged on a ChemiDoc Gel Imaging System (BioRad). Blots were quantified using ImageJ.

**Real-time RT-PCR.** MIN6 β cells were washed twice with ice-cold PBS and harvested using TRIzol (Invitrogen). Chloroform was added to the harvested samples, followed by centrifugation (12,000 rpm at 4 °C for 15 min) to separate aqueous and organic phases. The supernatant containing RNA was collected, and isopropanol was added to precipitate the RNA. Precipitated RNAs were pelleted and washed with 70% ethanol. RNA pellets were dried at room temperature for 10 min and dissolved in nuclease-free water. Recovered RNAs were quantified by NanoDrop 2000 (Thermo Scientific). cDNA was synthesized using PrimeScript RT Master Mix (TaKaRa) under conditions recommended by the manufacturer. A total of 40 ng of cDNA was used per reaction and amplified using iTaq Universal SYBR Green Supermix (BioRad) on a CFX96 Touch Real-Time PCR Detection System (Bio-Rad). Gene expression levels were calculated using the $2^{-\Delta\Delta Ct}$ method after normalizing to the expression level of β-Actin.

**Cell proliferation assays.** For stable MIN6 β cell lines, 1,000,000 cells were seeded in 6-cm dishes. The cell number was quantified using a Countess II cell counter (Life Technologies), with cells split every 5 days. Transiently transfected MIN6 β cells were seeded at a density of 30,0000 cells/well in 6-well plates and cell numbers counted once every two days for 8 days.

**Bioluminescence resonance energy transfer (BRET) assays.** White-walled, clear-bottom 96-well assay plates (Costar) were coated with 0.1 mg ml$^{-1}$ of poly-D-lysine (PDL) in Dulbecco's PBS (pH 7–7.3) for 24 h. HEK293T cells were transfected using PolyJet in 6-well plates (Costar) for 24 h and seeded at a density of $5 \times 10^4$ cells per well in the PDL-coated 96-well assay plates. Luminescence intensities were recorded on a Spark 20M fluorescence microplate reader using SparkControl Magellan 1.2 software (TECAN). Cells were washed twice with Hank's balanced salt solution (HBSS) and placed in 100 μL of HBSS containing 5 μM CTZ400a (DeepBlueC, Nanolight Technology) immediately prior to recording for eight cycles to obtain a baseline reading. To monitor R:C complex dissociation, wells received an additional 100 μL of CTZ400a-HBSS solution without (control) or with 100 μM Fsk (Calbiochem; final concentration 50 μM) and were read for an additional 15 cycles. For R subunit dimerization assays, wells were read for 10 cycles after receiving only 100 μL of HBSS containing 5 μM CTZ400a. Intensities were recorded from each well using the monochromator (400 ± 40 nm and 540 ± 35 nm) with a gain of 25 and a 1-s integration time. The entire plate was read one wavelength at a time, resulting in a time interval of ~ 2 min per cycle.

**Widefield epifluorescence imaging.** Puncta formation assays and RCaMP, ICUE4, cAMPFIRE, and PDE4D2$_{cat}$-ICUE4 biosensor imaging experiments were performed on an Ziess AxioObserver Z1 microscope (Carl Zeiss) with equipped with Definite Focus (Carl Zeiss), a 40×/1.4 NA oil-immersion objective, and a Photometrics Evolve EMCCD (Photometrics) and controlled by METAFLUOR 7.7 software (Molecular Devices). GFP intensities were imaged using a 480DF30 excitation filter, 505DRLP dichroic mirror, and 535F45 emission filter. RFP intensities were imaged using a 555DF25 excitation filter, 568DRLP dichroic mirror, and 650DF100 emission filter. Dual cyan/yellow emission ratio imaging was performed using a 420DF20 excitation filter, 455DRLP dichroic mirror, and two emission filters (475DF40 for CFP and 535DF25 for YFP). All filter sets were alternated using a Lambda 10-3 filter changer (Sutter Instruments). Exposure times for each channel were 500 ms, except YFP, 50 ms, with electron multiplying gain set to 50. Images were acquired every 20–30 s.

**Spinning-disk confocal imaging.** Additional puncta formation assays and colocalization assays were performed on a Nikon Ti2 microscope (Nikon) equipped with a Yokogawa W1 confocal scanhead (Yokogawa), an Opti Microscan FRAP unit (Bruker Nano Inc, integrated by Nikon), a six-line (405, 445, 488, 515, 561, and 640 nm) LUN-F-XL laser engine with bandpass and long-pass filters (450/50 and 525/50) (Chroma), an Apo TIRF 100×/1.49 NA objective, and a Prime95B sCMOS camera (Photometrics), operated with NIS Elements software (Nikon). Z-stacks with a 0.2-μm step length were obtained sequentially in each channel with 200 ms excitation at 20% laser power (405, 488, and 561 nm) every 1–2 min as indicated.

**Fluorescence recovery after photobleaching (FRAP).** MIN6 β cells were imaged on a Nikon Ti2 microscope (Nikon) equipped with a Yokogawa W1 confocal scanhead (Yokogawa), an Opti Microscan FRAP unit (Bruker Nano Inc, integrated by Nikon), a six-line (405, 445, 488, 515, 561, and 640 nm) LUN-F-XL laser engine with bandpass and long-pass filters (450/50 and 525/50) (Chroma), an Apo TIRF 100×/1.49 NA objective, and a Prime95B sCMOS camera (Photometrics) operated with NIS Elements software (Nikon). Circular regions of interest were drawn over whole puncta (approximately 0.30–1.00 μm radius) or similarly sized regions of bulk cytosol for photobleaching. Each experiment contained a minimum of 10 images at 0.5-sec intervals with 200 ms excitation using the 488 nm laser line at 20% power, then puncta or bulk cytosol regions of interest were bleached once using the 405 nm laser line for 500 ms at 75% power and recovery monitored every 0.5 s for a total of 2 min.

**In vitro droplet assay.** Protein solutions were prepared in buffer containing 150 mM KCl, 5 mM MgCl$_2$, 20 mM HEPES (pH 7.0), 1 mM EGTA, 1 mM DTT, and 0.5 mM ATP, with specified concentrations of cAMP and PEG 4,000. Following protein quantification via BCA assay, samples were prepared by combining purified proteins at various molar ratios and concentrations. The mixtures were incubated in glass-bottom, black-walled 96-well plates at room temperature for 30–60 min before examination under DIC microscopy.

**Insulin secretion assay.** MIN6-6 cells were seeded at $5 \times 10^5$ cells per well in 12-well plates. After seeding, cells were transfected with 200 ng of DNA and cultured for 48 h under standard conditions. Following the incubation period, cells were washed twice with PBS and starved for 1 hour in Krebs-Ringer Bicarbonate (KRB) buffer containing 114 mM NaCl, 5 mM KCl, 1.2 mM MgSO$_4$, 1.2 mM KH$_2$PO$_4$, 20 mM HEPES, 2.5 mM CaCl$_2$, 25.5 mM NaHCO$_3$, and 0.2% BSA. After starvation, cells were stimulated with 25 mM glucose and/or 1 μM GLP-1 for 1 h. For secreted protein analysis, culture media was collected and centrifuged at 2,000 rpm for 5 min to remove cellular debris. Then, 10 μL of the cleared supernatant was mixed with 5 μM CTZ400A (Nanolight Technology) for subsequent analysis. For total protein analysis, cells were lysed in RIPA buffer and centrifuged at 12,000 rpm for 5 min at 4 °C. The insulin-containing supernatant values were normalized to total cell lysate signal.

## Quantification and statistical analysis

**BRET biosensor analysis.** Emission ratios were calculated as $R = \frac{GFP^2}{RLuc8} - c.f.$ for each condition, with the control factor $(c.f.) = \frac{GFP2}{RLuc8}$ when only RLuc8 is present. Fsk-induced ratio changes were calculated as

$$\frac{\Delta R_{Fsk}}{\Delta R_0} = \frac{Avg.\ Control\ BRET\ Ratio - Avg.\ Fsk\ BRET}{Avg.\ Control\ BRET\ Ratio}.$$ Graphs were plotted using Prism 10 (GraphPad).

**FRET biosensor analysis.** Raw fluorescence images were corrected by subtracting the background intensity from an ROI without cells. The background-subtracted intensities of biosensor-expressing cells were used to calculate raw emission ratios (C/Y) at each time point ($R$). Time courses for cAMP compartmentalization assays were plotted as the normalized ratio change $\Delta R/\Delta R_{max}$, calculated as $\frac{R - R_{min}}{R_{max} - R_{min}}$, where $R_{min}$ is defined as the minimum $R$ across the time-course and $R_{max}$ is defined as the maximum $R$ across the time-course. For Ca$^{2+}$ oscillation time-courses, the background-subtracted fluorescence intensity ($F$) at each time point was normalized to the intensity at time $t = 0$ ($F/F_0$), which is defined as the time point immediately before drug addition. For some time courses, baseline slopes were corrected to a slope of 0 using $y = mx + b$ to offset a drifting baseline. Graphs were plotted using Prism 10 (GraphPad).

**Cellular puncta quantification.** Cell selection for analysis was determined by evaluating the raw confocal Z-stack images from each channel to identify cells with sufficient fluorescence intensity (i.e., expression) in each channel, as well as the stereotypical expression pattern and cell morphology. Z-stacks were converted to max intensity projections using Fiji (ImageJ). The number of puncta per cell was determined using the "find maxima" function within cellular ROIs. Prominence was adjusted depending on expression to isolate the puncta and record the number of maxima. Graphs were plotted using Prism 10 (GraphPad).

**FRAP analysis.** Intensity counts for puncta and unbleached reference puncta within the same cells were background subtracted using a nearby cell-free background ROI and normalized to the pre-bleach average intensity to account for any ordinary photobleaching over the experimental time-course. Then, corrected recovery values were normalized to unbleached reference puncta to correct for any effects of the FRAP laser. Finally, corrected recovery values for each bleached punctum were normalized to the bleached punctum time course's minimum and maximum intensities to calculate the fractional recovery on a scale of 0–1, with 1 indicating complete recovery to baseline intensity values. The normalized recovery curves were fitted to an exponential recovery curve using the ImageJ curve-fitting tool. Samples with bleached or reference puncta that moved out of the Z-plane during acquisition were truncated (after reaching steady-state) or excluded from analysis. Following curve-fitting, samples with curve y-intercepts $\geq 0.15$ were excluded from further analysis, since intensity values were normalized to 0 at $t=0$, the time of photobleaching. Time to half-maximum ($t_{1/2}$) values for each sample were derived from the ImageJ curve-fitting tool using $t_{\frac{1}{2}} = \frac{\ln_{f_0}(0.5)}{-\tau}$, where $\tau = b$ in the exponential recovery equation $y = a(1-e)(-bx) + c$. Apparent diffusion coefficients were calculated using the Soumpasis equation [39,40], $D_{app} = 0.224 \frac{r^2}{t_{1/2}}$. Graphs were plotted using Prism 10 (GraphPad).

**Colocalization analysis.** Line scans were performed in Fiji (ImageJ) to determine colocalization. Emission intensities ($I$) were normalized by dividing the intensity at each position by the maximum intensity ($I_{max}$) for reported line-scans ($I/I_{max}$). Graphs were plotted using Prism 10 (GraphPad).

**Statistics and reproducibility.** Statistical analyses were performed in GraphPad Prism 10 (GraphPad). For data sets with a normal distribution, pairwise comparisons were performed using unpaired Student's t-tests, with Welch's correction applied where indicated. Comparisons between more than two groups were performed using ordinary one-way analysis of variance (ANOVA) or Welch's ANOVA, followed by the indicated multiple-comparison test. For non-Gaussian (i.e., puncta quantification) data, pairwise comparisons before and after stimulation were performed using paired Wilcoxon signed-rank tests, and comparisons to WT were performed using unpaired Kolmogorov–Smirnov tests. Statistical significance was set at $P < 0.05$. Throughout the paper, datasets are combined from at least three independent experiments, except for puncta-per-cell data and in vitro droplet assay, which were combined from at least two independent experiments. Line-scans shown in S3B Fig are representative of at least 10 cells from three independent experiments.

## Supporting information

**S1 Fig. RIα LLPS Validation in MIN6 β Cells. (A)** Western blot analysis confirming successful CRISPR-mediated homology-directed repair insertion of GFP2 into the RIα locus in MIN6 β cells (KI). Asterisk indicates GFP2-tagged RIα. **(B)** Representative time courses of normalized recovery of fluorescence intensity after photobleaching of the indicated regions. **(C)** Quantification of the recovery half-time after photobleaching of stimulated bulk cytosol ($n = 44$ regions), stimulated puncta ($n = 41$ puncta), and basal large puncta ($n = 48$ puncta). ****$P = 5.89 \times 10^{-5}$, stimulated bulk cytosol versus stimulated puncta; ****$P = 1.80 \times 10^{-8}$, stimulated bulk cytosol versus basal large puncta; and ****$P = 1.05 \times 10^{-7}$, stimulated puncta versus basal large puncta; Brown-Forsythe and Welch one-way ANOVA with Dunnett's T3 multiple comparisons test. Error bars indicate mean ± SEM. **(D)** Quantification of apparent diffusion coefficients (Dapp) across different cellular compartments. The graph shows $\text{Log}_{10}$(Dapp) values in $\mu m^2$/s for stimulated bulk cytosol ($n = 68$), stimulated puncta ($n = 42$), and basal large puncta ($n = 48$). Statistical significance is indicated by asterisks (****$P < 0.0001$, ***$P < 0.001$).

Error bars indicate mean ± 95% CI, and statistics indicated by Brown-Forsythe and Welch ANOVA followed by Dunnett's T3 multiple comparisons test. The data underlying this figure can be found in S1 Data and S1 Raw Images.
(TIF)

**S2 Fig. Analysis of PKA subunit expression in RIα knockout MIN6 β cells, effects of RIα<sup>WT</sup> or RIα<sup>Y122A</sup> rescue, and schematic overview of the cAMP compartmentalization assay. (A)** Western blot results showing knockout of RIα in CRISPR/Cas9-engineered MIN6 β cell line. **(B)** Western blot analysis of PKA subunit expression in WT and RIα-null MIN6 β cells (left), and in RIα-null cells expressing either RIα<sup>WT</sup> or RIα<sup>Y122A</sup> (right). Representative blots from three independent experiments. (C) Quantification of western blot data comparing WT versus RIα-null MIN6 β cells (top) and RIα<sup>WT</sup> versus RIα<sup>Y122A</sup> expression (bottom). $n = 3$ experiments, ****$P < 0.0001$, unpaired two-tailed Student $t$ test. Data represent mean ± SD. **(D)** Schematic overview of the cAMP compartmentalization assay. The FRET-based cAMP indicator ICUE4 is tethered to the PDE4D2$_{cat}$ to specifically probe cAMP accumulation in the vicinity of PDE4D2. The data underlying this figure can be found in S1 Data and S1 Raw Images.
(TIF)

**S3 Fig. RIα<sup>Y122A</sup> disrupts cAMP-stimulated RIα LLPS. (A)** Representative maximum intensity projections from confocal Z-stacks showing GFP expression **(left)** and quantification of puncta per cell **(right)** from HEK293T cells co-expressing Cα-mCherry plus GFP2-tagged RIα<sup>WT</sup> (WT, $n = 90$ cells) or RIα<sup>Y122A</sup> (Y122A, $n = 59$ cells) before (−) and after (+) Fsk and IBMX stimulation. ****$P < 1 \times 10^{-15}$; 2-way ANOVA with Šídák's multiple comparisons test. **(B)** Representative fluorescence images and line-intensity profiles of the indicated regions in HEK293T cells expressing smAKAP-EGFP with mRuby2-tagged RIα<sup>WT</sup> or RIα<sup>Y122A</sup>. **(C)** BRET assay time-course **(left)** and quantification of Fsk-stimulated change **(right)** in the GFP2/RLuc8 emission ratio in HEK293T cells co-expressing GFP2-Cα plus RLuc8 fused to either RIα<sup>WT</sup> ($n = 3$ experiments) or RIα<sup>Y122A</sup> ($n = 3$ experiments). Untethered RLuc8 co-expressed with GFP2-Cα is used as a control. $P = 0.61$, unpaired, two-tailed Student $t$ test. Solid lines indicate the mean, and shaded areas indicate SEM. **(D)** Critical saturation concentration (C-sat) curves of RIα<sup>Y122A</sup>. In vitro phase diagram of RIα<sup>Y122A</sup> vs. PEG 4000 **(Left)**. Filled circles represent droplets, and empty circles represent no droplets. C-sat curves showing PKA-Cα concentration-dependent LLPS behavior of RIα<sup>WT</sup> (green) and RIα<sup>Y122A</sup> (magenta) **(Right)**. Dotted lines indicate the boundary between one-phase and two-phase regions (shaded areas). RIα<sup>WT</sup> data were modified from our previous work [10]. **(E)** Quantification showing basal puncta number per cell in RIα-null MIN6 β cells expressing either mRuby2-tagged RIα<sup>WT</sup> (WT; $n = 98$ cells) or RIα<sup>Y122A</sup> (Y122A; $n = 103$ cells). ****$P = 5.47 \times 10^{-12}$ (WT vs. Y122A); unpaired Komogorov-Smirnov test. Error bars in summary quantification indicate median ± 95% CI. **(F)** Representative images **(left)** and quantification of RIα<sup>Y122A</sup> puncta number ($n = 53$ cells) **(middle)** and area (for pre-existing puncta) ($n = 15$ puncta) **(right)** in RIα-null MIN6 β cells. ns, $P = 0.17$ for puncta number per cell; ns, $P = 0.12$ for pre-existing puncta area; paired, two-tailed Student's $t$ tests. All scale bars, 10 μm. The data underlying this figure can be found in S1 Data.
(TIF)

**S4 Fig. Ca²⁺ and cAMP oscillations of RIα-null MIN6 β cells. (A)** Heatmap **(left)** showing Ca²⁺ oscillations in individual WT or RIα-null (RIα KO) MIN6 β cells (each row) plotted as normalized RCaMP intensity. Cells were ordered by the period of the first two Ca²⁺ peaks after TEA stimulation. Representative single-cell traces of Ca²⁺ oscillations **(top right)** and oscillation period **(bottom right)** in RIα KO ($n = 61$ cells) and WT ($n = 53$ cells) MIN6 β cells. ***$P = 4 \times 10^{-4}$; unpaired, two-tailed Student $t$ test. **(B)** Heatmap **(left)** depicting temporal dynamics of cAMP oscillations in individual WT or RIα-null (RIα KO) MIN6 β cells (each row), shown as normalized cAMPFIRE C/Y emission ratios. Cell ordering reflects the period between the initial two cAMP peaks following TEA stimulation. Representative cAMP oscillation traces **(top right)** and period quantification **(bottom right)** in WT ($n = 19$ cells) or KO ($n = 31$ cells). ***$P = 0.0002$; unpaired, two-tailed Student $t$ test. Data represent mean ± SEM. The data underlying this figure can be found in S1 Data.
(TIF)

**S5 Fig. Insulin secretion assay using MIN6-6. (A)** Insulin secretion assay in MIN6-6 cells under different stimulation conditions. Cells were treated with glucose (25 mM), GLP-1 (1 μM), or both. (Glucose: ****$P < 0.0001$; GLP-1: **$P < 0.0052$; Glucose and GLP-1: ****$P < 0.0001$; $n = 3$ for all conditions). Statistical analysis was performed using ordinary one-way ANOVA followed by Dunnett's multiple comparisons test. Error bars represent mean ± SEM. **(B)** Western blot results showing knockout of RIα in MIN6-6 β cell line. **(C)** Relative insulin secretion level between parent MIN6-6 and RIα-null MIN6-6 cells upon glucose (25 mM)/GLP-1(1 μM) stimulation. $n = 3$ for both, ***$P = 0.000106$; unpaired, two-tailed Student $t$ test. Error bars indicate mean ± SEM. **(D)** Representative fluorescence images **(left)** and quantification of RIα$^{K123A}$ puncta number per cell before (−) and after (+) Fsk (50 μM) and IBMX (100 μM) stimulation. ($n = 14$ cells, ***$P = 0.0005$) **(right)**. **(E)** Relative insulin secretion levels in RIα-null MIN6-β cells expressing RIα$^{WT}$, RIα$^{Y122A}$, or RIα$^{K123A}$ ($n = 3$). *$P = 0.0415$, **$P = 0.0078$, ***$P = 0.0005$; ordinary one-way ANOVA followed by Tukey's multiple-comparisons test. Error bars indicate mean ± SEM. The data underlying this figure can be found in S1 Data and S1 Raw Images.
(TIF)

**S6 Fig. Cellular function of RIα-null MIN6 β cells. (A)** Western blot of CREB and phospho-CREB **(top)** and quantification of p-CREB/CREB **(bottom)** in WT and RIα-null (KO) MIN6 β cells ($n = 4$ experiments). **$P = 2.62 \times 10^{-3}$; unpaired, two-tailed Student $t$ test. **(B)** Quantification of Cyr61 and JunB mRNA levels in RIα KO MIN6 β cells ($n = 7$ experiments). ****$P = 5.9 \times 10^{-8}$; **$P = 6.68 \times 10^{-4}$; unpaired, two-tailed Student $t$ test. **(C)** Cell proliferation of RIα-null (RIα KO) and wild-type (WT) MIN6 β cells ($n = 4$ experiments). **$P = 1.82 \times 10^{-3}$; Multiple unpaired $t$ test. Error bars indicate mean ± SEM. The data underlying this figure can be found in S1 Data and S1 Raw Images
(TIF)

**S1 Movie. Dynamics of RIα phase-separated bodies in MIN6 β cells after Fsk/IBMX stimulation, related to Fig 1.** Live-cell imaging of RIα-GFP2 in MIN6 β cells co-expressing RIα-GFP2 and TagBFP2-PKA-Cα following stimulation with Fsk/IBMX. The arrowhead points to two puncta that merge over the course of the movie.
(AVI)

**S2 Movie. Live-cell imaging of RIα$^{Y122A}$ in MIN6 β cells after Fsk/IBMX stimulation, related to S2 Fig.** Live-cell imaging of RIα$^{Y122A}$-GFP2 in MIN6 β cells co-expressing RIα$^{Y122A}$-GFP2 and TagBFP2-PKA-Cα following stimulation with Fsk/IBMX.
(AVI)

**S1 Raw Images. Unedited Western blot images.**
(PDF)

**S1 Data. Raw data associated with the manuscript.**
(XLSX)

**S1 Table. Oligonucleotides for RT-PCR, cloning, and CRISPR/Cas9 based cell engineering.**
(DOCX)

## Acknowledgments

We thank Eric Griffis and Peng Guo from the Nikon Imaging Center at UCSD for their training and assistance with the spinning-disk confocal microscope. We thank Dr. Jun-Ichi Miyazaki from Osaka University for gifting the MIN6 β cells, Dr. Wenbiao Chen from Vanderbilt University for sharing the MIN6-6 cells, Dr. Loren Looger from the University of California San Diego for gifting the RCaMP, and Dr. Haining Zhong from Oregon Health & Science University for providing the cAMPFIRE used in this paper.

## Author contributions

**Conceptualization:** Ha Neul Lee, Julia C. Hardy, Sohum Mehta, Jin Zhang.

**Data curation:** Ha Neul Lee, Julia C. Hardy.

**Formal analysis:** Ha Neul Lee, Julia C. Hardy, Jin-Fan Zhang, Su Hyun Kim.

**Funding acquisition:** Ha Neul Lee, Jin Zhang.

**Investigation:** Ha Neul Lee, Julia C. Hardy, Jin-Fan Zhang, Su Hyun Kim, William F. Buhl, Jessica G.H. Bruystens, Sohum Mehta, Susan S. Taylor, Jin Zhang.

**Methodology:** Ha Neul Lee, Julia C. Hardy, Emily H. Pool, Jin-Fan Zhang, Su Hyun Kim, William F. Buhl, Jessica G.H. Bruystens, Sohum Mehta.

**Project administration:** Ha Neul Lee, Julia C. Hardy, Jin Zhang.

**Resources:** Ha Neul Lee, Julia C. Hardy, Jin Zhang.

**Software:** Julia C. Hardy.

**Supervision:** Ha Neul Lee, Julia C. Hardy, Susan S. Taylor, Jin Zhang.

**Validation:** Julia C. Hardy, Emily H. Pool, Jin-Fan Zhang.

**Visualization:** Julia C. Hardy, Emily H. Pool.

**Writing – original draft:** Ha Neul Lee, Julia C. Hardy, Emily H. Pool, Sohum Mehta, Jin Zhang.

**Writing – review & editing:** Ha Neul Lee, Julia C. Hardy, Emily H. Pool, Jin-Fan Zhang, Su Hyun Kim, William F. Buhl, Jessica G.H. Bruystens, Sohum Mehta, Susan S. Taylor, Jin Zhang.

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
