## [Editor Report · Decision Letter 0]

Dear Dr Zhang,

Thank you for submitting a revised version of your manuscript entitled "PKA RIα Phase Separation Regulates β Cell Function through cAMP Compartmentation" for consideration as a Short Report by PLOS Biology.

Your manuscript has now been evaluated by the PLOS Biology editorial staff and I am writing to let you know that we would like to send your submission back to the original reviewers for their feedback on the revision.

Once your full submission is complete, your paper will undergo a series of checks in preparation for peer review. After your manuscript has passed the checks it will be sent out for review. To provide the metadata for your submission, please Login to Editorial Manager (https://www.editorialmanager.com/pbiology) within two working days, i.e. by Apr 09 2025 11:59PM.

Kind regards,

Luke

Lucas Smith, Ph.D.

Senior Editor

PLOS Biology

lsmith@plos.org

---

## [Decision Letter · Decision Letter 1]

Dear Dr Zhang,

Thank you for your patience while we considered your revised manuscript "PKA RIα Phase Separation Regulates β Cell Function through cAMP Compartmentation" for publication as a Short Report at PLOS Biology. This revised version of your manuscript has been evaluated by the PLOS Biology editors, the Academic Editor and two of the original reviewers, who have suggested we accept the study.

Based on the reviewers' assessment, we are likely to accept your manuscript. However before we can editorially accept your study we would like to invite you to address a few editorial, data, and policy-related requests in a short revision. These are detailed below.

*IMPORTANT: Please address the following editorial requests:

1) TITLE: We would like to suggest a minor modification to the title. If you agree we suggest the title be changed to:

"Phase separation of the PKA type I regulatory subunit regulates β-cell function through cAMP compartmentalization"

(we are suggesting this change, as we think it may not be immediately obvious to our broad readership, what R1a is).

2) Also, we noticed that you used the word 'compartmentation' in the title and throughout the paper, and wonder if this should be changed to "compartmentalization". A quick google search suggests that these can be used interchangeably, so if you have a strong preference for compartmentation, we are OK leaving it as is. But we think compartmentalization is used more commonly in this context.

3) FINANCIAL DISCLOSURES: Please update your financial disclosures statement to indicate whether the sponsors or funders played any role in the study design, data collection and analysis, decision to publish, or preparation of the manuscript.

4) DATA AVAILABILITY: You may be aware of the PLOS Data Policy, which requires that all data be made available without restriction: http://journals.plos.org/plosbiology/s/data-availability. For more information, please also see this editorial: http://dx.doi.org/10.1371/journal.pbio.1001797

a. Supplementary files (e.g., excel). Please ensure that all data files are uploaded as 'Supporting Information' and are invariably referred to (in the manuscript, figure legends, and the Description field when uploading your files) using the following format verbatim: S1 Data, S2 Data, etc. Multiple panels of a single or even several figures can be included as multiple sheets in one excel file that is saved using exactly the following convention: S1_Data.xlsx (using an underscore).

b. Deposition in a publicly available repository. Please also provide the accession code or a reviewer link so that we may view your data before publication.

>>Regardless of the method selected, please ensure that you provide the individual numerical values that underlie the summary data displayed in the following figure panels as they are essential for readers to assess your analysis and to reproduce it:

Fig. 1B-C,E,G; Fig 2A-F; Fig 3A-F;

Fig S1B-D; Fig S2C; Fig S3A-F; Fig S4A-B; Fig S5A,C,E; Fig S6A-C;

>>Please also ensure that figure legends in your manuscript include information on where the underlying data can be found, and ensure your supplemental data file/s has a legend.

>>Please ensure that your Data Statement in the submission system accurately describes where your data can be found.

5) BLOT AND GEL REPORTING: We require the original, uncropped and minimally adjusted images supporting all blot and gel results reported in an article's figures or Supporting Information files. We will require these files before a manuscript can be accepted so please prepare and upload them now. Please carefully read our guidelines for how to prepare and upload this data: https://journals.plos.org/plosbiology/s/figures#loc-blot-and-gel-reporting-requirements

6) CODE: Per journal policy, if you have generated any custom code during the course of this investigation, please make it available without restrictions. Please ensure that the code is sufficiently well documented and reusable, and that your Data Statement in the Editorial Manager submission system accurately describes where your code can be found.

We expect to receive your revised manuscript within two weeks.

*Published Peer Review History*

*Press*

Sincerely,

Luke

Lucas Smith, Ph.D.

Senior Editor

lsmith@plos.org

PLOS Biology

Reviewer remarks:

Reviewer #2: The authors should be commended for their thorough revision that significantly has improved the manuscript. They have dealt satisfactorily with all comments from this reviewer.

Reviewer #3: The authors have thoroughly addressed my previous concerns, and I am satisfied with the revisions made to the manuscript

---

## [Editor Report · Decision Letter 2]

Dear Dr Zhang,

Thank you for the submission of your revised Short Report "Phase separation of a PKA type I regulatory subunit regulates β-cell function through cAMP compartmentalization" for publication in PLOS Biology and thank you for addressing our last editorial requests in this revision. On behalf of my colleagues and the Academic Editor, Susanne Mandrup, I am pleased to say that we can in principle accept your manuscript for publication, provided you address any remaining formatting and reporting issues. These will be detailed in an email you should receive within 2-3 business days from our colleagues in the journal operations team; no action is required from you until then. Please note that we will not be able to formally accept your manuscript and schedule it for publication until you have completed any requested changes.

**IMPORTANT: Thank you for providing me with an updated file containing the uncropped western blot images related to your study. I have replaced the previous version of this, with the updated file that you provided me over email. Please take a moment to double check that everything looks OK after this change.

PRESS

Sincerely, 

Lucas Smith, Ph.D.

Senior Editor

PLOS Biology

lsmith@plos.org